# iSCNT embryo culture system for restoration of *Cervus nippon hortulorum*, presumed to be sika deer in the Korean Peninsula

Yong-Su Park[1☯], Min-Gee Oh[2☯], Sang-Hwan Kim [ID][2,3,4]*

**1** National Institute of Ecology, Research Center for Endangered Species, Seocheon-gun, Chungcheongnam-do, Korea, **2** General Graduate School of Animal life convergence science, Hankyong National University, Ansung, Gyeonggi-do, Republic of Korea, **3** School of Animal Life Convergence Science, Hankyong National University, Ansung, Gyeonggi-do, Republic of Korea, **4** Institute of Applied Humanimal Science, Hankyong National University, Unsung, Ansung, Gyeonggi-do, Republic of Korea

☯ These authors contributed equally to this work.
* immunoking@hknu.ac.kr

**Data Availability Statement:** All relevant data are within the manuscript and its Supporting Information files.

**Funding:** "The authors received no specific funding for this work."

## Abstract

Sika deer inhabiting South Korea became extinct when the last individual was captured on Jeju Island in Korea in 1920 owing to the Japanese seawater relief business, but it is believed that the same subspecies (Cervus nippon hortulorum) inhabits North Korea and the Russian Primorskaya state. In our study, mt-DNA was used to analyze the genetic resources of sika deer in the vicinity of the Korean Peninsula to restore the extinct species of continental deer on the Korean Peninsula. In addition, iSCNT was performed using cells to analyze the potential for restoration of extinct species. The somatic cells of sika deer came from tissues of individuals presumed to be Korean Peninsula sika deer inhabiting the neighboring areas of the Primorskaya state and North Korea. After sequencing 5 deer samples through mt-DNA isolation and PCR, BLAST analysis showed high matching rates for Cervus nippon hortulorum. This shows that the sika deer found near the Russian Primorsky Territory, inhabiting the region adjacent to the Korean Peninsula, can be classified as a subspecies of Cervus nippon hortulorum. The method for producing cloned embryos for species restoration confirmed that iSCNT-embryos developed smoothly when using porcine oocytes. In addition, the stimulation of endometrial cells and progesterone in the IVC system expanded the blastocyst cavity and enabled stable development of energy metabolism and morphological changes in the blastocyst. Our results confirmed that the individual presumed to be a continental deer in the Korean Peninsula had the same genotype as Cervus nippon hortulorum, and securing the individual's cell-line could restore the species through replication and produce a stable iSCNT embryo.

## Introduction

The study of the phylogeny of *Cervidae* has been ongoing for a very long time, and it is thought to have evolved from the development of Pecorans (a superordinate clade of Cervidae) in the Mid-Eocene. Several recent studies have classified several species under the genus *Cervus*. *Elaphurus davidianus*, *Przewalskium albirostris*, *Rucervus eldii*, some Russian species, and

**Competing interests:** The authors have declared that no competing interests exist.

occasionally *Panolia eldii*, have been incorrectly included in the genus *Rucervus*. The analysis also revealed a significant evolutionary and systematic distance between *Panolia eldii* and *Rucervus duvaucelii*. In particular, the sika deer distributed in the Korean Peninsula are *Cervus nippon hortulorum*, which inhabits southern China and the Russian Far East, and seems to have been distributed throughout the Korean Peninsula [1]. Sika deer inhabiting South Korea became extinct when the last individual was captured on Jeju Island in Korea in 1920 owing to the Japanese seawater relief business, but it is believed that the same subspecies (*Cervus nippon hortulorum*) inhabits North Korea and the Russian Primorskaya state.

Later, several species were flowed into the Korean Peninsula for individual restoration, but the somatic cell classification of *Cervus nippon hortulorum*, which is considered endemic to the Korean Peninsula, proposed by Whitehead [1], was insufficient. In addition, it is officially extinct in South Korea owing to environmental and biological problems. Furthermore, it is challenging to introduce individuals which may carry diseases such as foot-and-mouth disease. *Cervus* is thought to be closely related to 6–7 taxa, including *Cervus timorensis*, *Cervus unicolor*, *Cervus albirostris*, *Cervus nippon*, *Cervus elaphus*, *Cervus canadensis*, and *Cervus eldii* [2,3]. However, some studies have shown differences in phylogenetic information between morphological and mitochondrial DNA marker classification methods; therefore, it is believed that species that flow into the Korean Peninsula are diverse.

It is beyond any doubt that the tremendously low efficiency of mammalian somatic cell nuclear transfer (SCNT)-mediated cloning, including especially its interspecies model, can be improved only by comprehensively recognizing molecular determinants and mechanisms affecting the developmental competencies of nuclear transfer-derived embryos [4,5]. The main impact on the development of cloned embryos is exerted by the type and source of nuclear donor cells [6,7]. In this context, an important role is also played by the strategies used to artificially synchronize the mitotic cycle of ex vivo-expanded nuclear donor cells at the G0/G1 stages [8–10]. It is also noteworthy that the developmental outcome of somatic cell-cloned embryos is largely determined by the molecular quality parameters reflected in the incidence of apoptotic cell death and oxidative stress processes in the in vitro cultured nuclear donor cells and SCNT-derived embryos [11,12]. Moreover, the developmental capability of cloned embryos is remarkably affected by the molecular quality of metaphase II-stage nuclear recipient oocytes, which depends largely on coordination between the processes of meiotic, cytoplasmic, and epigenomic maturation [13–15]. Finally, the effectiveness of generating somatic-cell-cloned embryos results highly from both epigenetic re-programmability of donor cell nuclei [16–18] and molecular interrelations between nuclear and mitochondrial genomes in developing SCNT-derived embryos [19–22].

Species restoration using interspecific somatic cell nuclear transfer (iSCNT) technology is thought to provide important information on the ecological adaptation of wild species and physiological research, such as the effect of the ecological environment of extinct species on individual habitats and species reproduction [23,24]. Hence, the most suitable iSCNT should be found using donor cells with high mitochondrial DNA consistency in somatic cells to restore species in the Korean Peninsula. Nuclear transfer, embryo security, and hatching blastocyst efficiency are essential for successful SCNT in the known interspecies. In the case of pigs, with many studies on interspecies SCNT, it is known that the formation rate of the inner cell mass (ICM) is high after SCNT, and cytoplasmic activity is also high after electrical stimulation [25,26]. In particular, blastocyst formation and hatching efficiency in pig SCNT is higher than in other interspecies nuclear transfer animals [27,28]. Therefore, this study was limited to Russia, Primorskaya state, and North Korea, where the same subspecies of deer inhabiting the Korean Peninsula was used for the domestic restoration of extinct sika deer, and somatic cells were obtained through research exchanges and MOUs with researchers from other countries.

Subsequently, somatic cells, presumed to be from Korean *Cervus nippon hortulorum*, were used to distinguish the species using a mitochondrial DNA marker [29], which was previously used to classify the genus *Cervus*. Furthermore, iSCNT conditions were established using somatic cells of Korean *Cervus nippon hortulorum* to determine the optimal IVC culture method that can affect blastocyst development, suggesting the possibility of cloning the Korean sika deer.

## Materials and methods

### Animal tissues

The somatic cells of sika deer came from tissues of individuals presumed to be Korean Peninsula sika deer inhabiting the neighboring areas of the Primorskaya state and North Korea, from the Primorskaya State Academy of Agriculture signed MOU(Date: 25.11.2014) in a previous study (Jangsu State of Korea, Project No.: 82-2014-003). All animal handling and experimental procedures followed a protocol approved by the Hankyong National University Animal Experimental Ethics Committee (IACUC approval HK-2021-1). Since the 1950s, blood from the descendants of sika deer, believed to be a species of Korean Peninsula, has been supported by the Jangsu County of Korea and used as a control group. The tissue and blood samples of sika deers used in this study are being stored at Hankyung National University, and the genetic information is currently registered in the NCBI (Repository: Institute of Animal Developmental Biotechnolo of Hankyong National University in Korea; Biobank: www.ncbi. nlm.nih.gov/bioproject/PRJNA831892; Organization: Hankyong National University of Korea). The sika deer tissues used in the experiment were named CnD22166 (Cervus nippon Deer: sample, tissue;), CnD 22119, CnD 11102, CnD 6, and CnD 8; and the control group was named WCnD (Wild Cervus nippon Deer of Jangsu-gun, Jeollabuk-do, Korea: blood; 2014). Porcine ovaries used as ISCNTs were collected from the prepubertal gilts at a local slaughterhouse.

### Extraction of mt-DNA

Mitochondrial DNA was extracted from a total of six deer samples following the manual method using a mitochondrial DNA isolation kit (cat no. K280-50, Biovision, Kor) from tissues and blood, and then dissolved in 20 μL TE buffer.

### PCR and sequencing

For amplification of the D-loop region of mitochondrial DNA, PCR was performed using the Anti HS taq -High- (cat no. AHS-101, TNT research, Kor), and Antk HS Taq premix (2x) (cat no. AHP-201, TNT research) PCR KIT with 20 ng/μL template DNA and 10 pmol primers (CST2; 5-TAATATACTGGTCTTGTAAACC-3 and CST39; 5-GGGTCGGAAGGCTGGGACCAAACC-3). The amplified PCR products were confirmed by electrophoresis on a 1.5% agarose gel. The PCR product showing a double band was purified to the desired length by Mag Extractor-PCR & Gel clean up (cat no. F0986K, TOYOBO, Jap) according to the manual. The purified PCR products were analyzed for the nucleotide sequence of the amplified product using an ABI Prism 3730xl DNA sequencer (Applied Biosystems, USA).

### Sequence analysis

Basic Local Alignment Search Tool (BLAST) was performed by selecting the nucleotide BLAST option from the BLAST website (http://blast.ncbi.nlm.nih.gov/Blast.cgi). The E-value is a result of 0, the five subject sequences linked above were selected, and the nucleotide

sequence and alignment were used as a query. The Molecular Evolutionary Genetics Analysis program (version 6.0) was used for phylogenetic analysis and classification.

### Preparation of donor cells

The donor cells were prepared by mincing the tissue of each sika deer. These were then cultured in Eagle's medium (DMEM with F-12, Invitrogen) supplemented with 10% fetal bovine serum (FBS, Invitrogen, USA), until the cells became a complete monolayer. The cells were left to grow for approximately 10–15 days. Cells were observed to grow every 2–3 days, and photographed for normal growth. Some cells were placed in a freezing vial of $1 \times 10^6$ cells in culture medium containing 10% dimethyl sulfoxide (DMSO; Sigma, USA), placed in a freezing tank (LN2:-196˚C) for 3–4 days, and stored in liquid nitrogen. Cells used for nuclear transfer were separated into 0.05% trypsin-EDTA (Invitrogen) after inducing the G0/G1 stage of the cell cycle for 72–96 h in a culture medium supplemented with 0.5% FBS by applying the method of Han et al. (2019). Dissociated donor cells were resuspended in HEPES-buffered Tyrode's medium (TLH, Invitrogen) supplemented with 0.4% (w/v) bovine serum albumin (BSA; Sigma-Aldrich) before nuclear transfer (Fig 1).

### Oocyte collection and in vitro maturation

Ovaries were collected from pigs slaughtered in slaughterhouses, maintained at a temperature of 37˚C in physiological saline solution containing antibiotics (penicillin G IU/mL, streptomycin 100 μg/mL), and transported to the laboratory within 2 h. The transferred ovaries were washed three times with saline, and follicle fluid and immature oocytes were collected from 3–5 mm follicles using a 10 mL syringe with an 18 gauge needle. The immature oocytes were washed three times with T-washing medium supplemented with 10 μg/mL antibiotic (Gibco, USA) and 0.3% (w:v) bovine serum albumin (BSA; Sigam, USA) in Hepes-buffered tissue culture medium 199 (TCM199; Gibco, USA), and primary step of in vitro maturation was

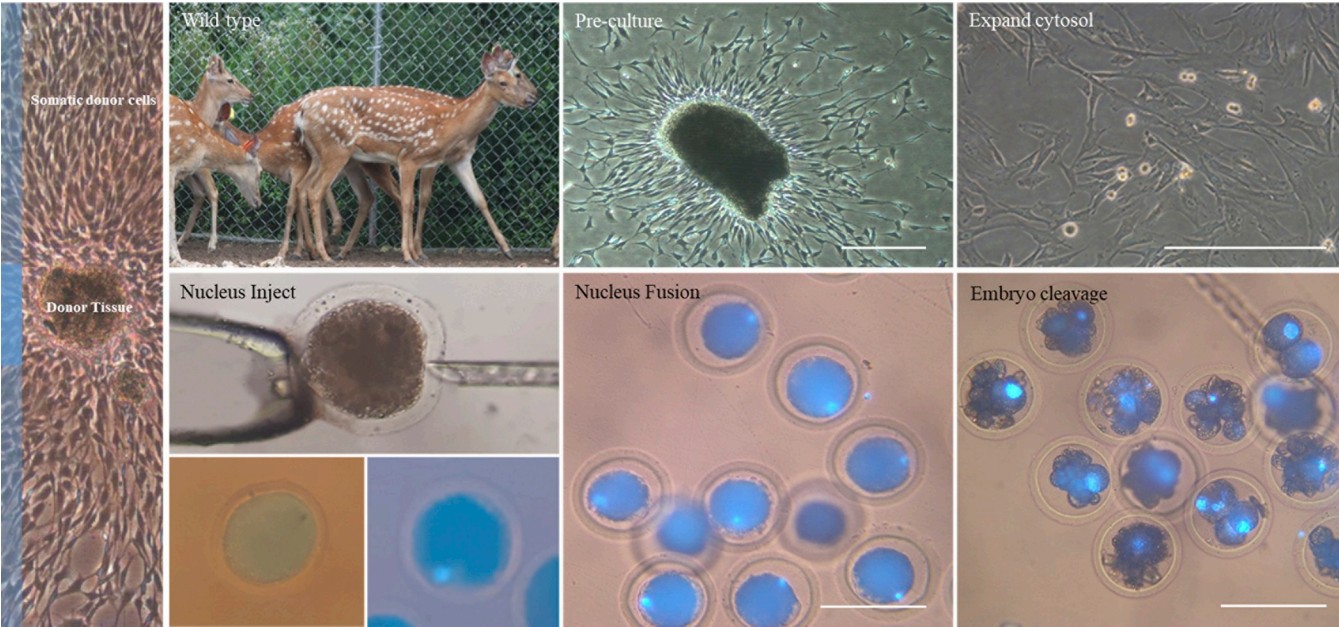

**Fig 1. Deer tissue culture and SCNT process.** The photographs above show the successive stages of culturing deer's ear tissue to produce a cell-line. The photographs below show generation of SCNT-derived embryos and their subsequent in vitro culture. (Scale bar = 100 μm).

induced on TCM199 maturation medium supplemented with 2.5 µg/mL gonadotropic hormone (GTH; Sigma, USA), 15 ng/mL epidermal growth factor (EGF; Sigma, USA), 30 µg/mL kanamycin (Sigma, USA) and 10% fetal bovine serum for 22 h in an incubator at 5% $CO_2$, 95% air, and 39˚C. The second step of in vitro maturation was induced by culturing for 22 h in the same in vitro medium, using a medium from which GTH was removed.

### Enucleation and nuclear transplantation

Pipettes used for nuclear transplantation were manufactured using a capillary tube with a diameter of 100 mm for denuclearization and injection. The outer diameters of the enucleation and injection pipettes were set at 20–30 µm. holding pipettes (RIGIO Inc., USA) with an outer diameter of 120–150 and an angle of 15˚ were used. The manufactured pipettes were washed with $H_2SO_3$ and distilled water, and sterilized before use. Oocytes induced to mature in vitro were placed in TCM-199 (Sigma-Aldrich) supplemented with 0.1% hyaluronidase (Sigma-Aldrich), vortexed for 4 min to remove cumulus cells, and washed with fresh calcium-free HEPES-buffered Tyrode's medium (TLH; Sigma-Aldrich) supplemented with 0.4% (w/v) BSA.

Then, the cells were transferred to a micromanipulation drop of calcium-free TLH supplemented with 0.4% (w/v) BSA and 7.5 µg/mL cytochalasin B (Sigma-Aldrich), and the nuclei were removed by suctioning the metaphase II (MII) chromosome with a 17 µm beveled pipette (Humagen, Charlottesville, VA) under a fluorescence microscope (IX70; Olympus, Tokyo, Japan). Somatic cell nuclear transfer was completed by injecting donor cells into the perivitelline space of the denuclearized oocyte.

### Fusion and activation

Electrical fusion of oocytes after nuclear transplantation was performed by applying the experimental method of Ryu and Yoon (2018) and energizing 1.1 kv/cm of direct current for 30 µsec in 280 Mm mannitol containing 0.1 mM $CaCl_2$, 0.1 mM $MgCl_2$, with calcium-free TLH by inducing cell fusion. Fused nuclear embryos were incubated for 4 h in a culture medium supplemented with 2 mM 6-dimethylaminopurine (6-DMAP; Sigma) and 7.5 µg/ml cytochalasin B to induce activation (Fig 1).

### In vitro culture

Activated iSCNT-derived embryos were cultured in NCSU-23 [30], PZM-3 [11], or PZM-5 [23] droplets covered with mineral oil (10–15 embryos/droplets). These were cultured for 7 days in an incubator with an environment of 5% $CO_2$, 5% $O_2$, and 90% $N_2$ at a temperature of 39˚C. For the co-culture method, the experimental method of Hatoya et al. (2006) was applied, and the medium selected in the in-vitro culture (IVC) method was used with cumulus cells (CC), oviduct epithelial cells (OEC), uterine endometrium cells (UEC), and iSCNT embryos. The co-culture was carried out in the same environment as the IVC culture environment. In particular, in the case of UEC, to evaluate the effects of hormones, luteinizing hormone (LH; Sigma-Aldrich), progesterone (P4; Sigma-Aldrich), and gonadotropin (GTH; Sigma-Aldrich) were mixed with 5 IU (v: v) of each, and iSCNT embryos were cultured.

### Statistical analysis

For statistical analysis of the results obtained in this study, the Tukey-Kramer multiple comparison test was performed using GraphPad Instat ver. 3.0. Statistical significance was defined as a difference of $p < 0.05$.

## Results

### Phylogenetic analysis of the mt-DNA D-loop region of Korean sika deer cells

As a result of analyzing the mt-DNA D-Loop region of the sika deer, CnD22166 has a total base sequence of 1,128 bp, CnD22119 of 1,132 bp, and CnD11102 of 1,115 bp among the cell lines (Table 1). The nucleotide sequence of the D-Loop in the blood of sika deer with WCnD was 1,100 bp, showing a difference within each nucleotide sequence. In addition, as a result of analyzing the nucleotide sequence matching of each gene in BLAST to confirm the dendrogram of the sika deer, the results showed high matching rates of CnD22166 with *Cervus nippon hortulorum*, CnD22119 with *Cervus hortulorum*, CnD11102 with *Cervus nippon hortulorum*, CnD6 with *Cervus hortulorum*, CnD8 with *Cervus hortulorum*, and WCnD with *Cervus nippon taiouanus*. In particular, CnD22166 showed an average of 99.9% nucleotide sequence identity, and among them, it was formed in the sequence of *Cervus nippon* isolate NIP2 tRNA, and it was confirmed that high identity was formed in the mt-DNA of *Cervus nippon hortulorum*. In the case of CnD11102, over 97% identity in the nucleotide sequence was measured, of which high identity was confirmed in the sequence of *Cevus nippon hortulorum* (Table 2). These results confirmed that two of the five highly evaluated individuals in the cell line of the sika deer showed very close genetic identification to *Cevus nippon hortulorum*, which is assumed by to be the sika deer species on the Korean Peninsula (Fig 2). However, the other three showed genetic differences from Cevus nippon hortulorum, making it difficult to regard them as native deer to the Korean Peninsula.

### Effect of culture medium on the developmental potential of iSCNT-derived hybrid embryos

Table 3 shows the development rate of interspecific iSCNT embryos in the culture medium. In the final blastocyst development of embryos, blastocysts were confirmed on day 7 of in vitro culture in NCSU-23 and PZM-5, which are commonly used for the development of pig embryos, and iSCNT was found to be high in NCSU-23. However, in PZM-3 medium, which is highly utilized as an amino acid culture medium, embryonic development was very low and did not develop beyond four cells. Early embryos showed a high cleavage rate in PZM-5 from the 2 to 8 cell stage, but development from morula to expansion of blastocysts was lower than that of NCSU-23. In particular, the blastocyst cavity was sufficiently expanded in the blastocysts of NCSU-23 compared to PZM-5 when comparing the morphological changes in the blastocysts on day 7 of in vitro culture (Fig 3A). Furthermore, the blastomere rate of the ICM section in NCSU-23 was increased, and the cell homogeneity of trophectoderm was shown to promote the morphological differentiation of blastocysts. However, there was no difference in the number of cells in the blastomeres (Fig 3B).

### Effect of co-culture type on the developmental potential of hybrid cloned embryos

To improve the embryonic cell division rate, the development efficiency of IVC of embryos was evaluated by adding hormones to the co-culture with CC, OEC, and UEC. Cleavage rates of the early embryos were high in all groups except UEC-LH, but the cleavage activities after the 4-cell stage were found to be significantly higher in the UEC-non and UEC-P4 groups. Moreover, blastocyst formation was observed in the UEC-non and UEC-P4 groups. It is also worth noting that the blastocyst formation rate was highest in the UEC-P4 group (Table 4). In the case of blastocyst cell homogeneity of UEC-non and UEC-P4 groups, the UEC-P4 group

**Table 1. Differences in the mt-DNA sequence of each sika deer sample.**

| | Mitochondria DNA D-Loop Sequencing | |
|---|---|---|
| CnD22J66 | TTA TTAATATAGTTCCATAAAAATCAAGAAACTTTATCAGTATTAAATTTCCAAAAAATTTTAATATTTTAATACAGCTTTCTACTTTATTAATATAGTTCCATAAAAATCAAGAACTTTATCAGTA | 172 |
| CnD22J19 | AACACCTTCCCTAGACTCA-GGAAGAAGCCATAGCCCCATTCTCATCACACCCAAAGCTGAACTTCTATTTAAACTATTTCCCTGACGC | |
| CnD11J02 | -CACCTCCCTAGACTCA-GGAAGAAGCCATAGCCCCACTATCAACACCCAAAGCTGAAGTTCTATTTAAACTATTTCCTGACGC | |
| CnDJ8 | -CACCTCCCTAGACTCAGGGAAGAAGCCATAGCCCCACTATCAACACCCAAAGCTGAAGTTCTATTTAAACTATTTCCTGACGC | |
| CnDJ6 | -CACCTCCCTAGACTCAGGGAAGAAGCCATAGCCCCACTATCAACACCCAAAGCTGAAGTTCTATTTAAACTATTTCCTGACGC | |
| WCnD | AACACCT-CCTAGACTCAGGGAAGAAGCCATAGCCCCACTATCATCACACCCAAAGCTGAAGGTTCCATTTAAACTATTTCCTGACGC | |
| 173 | (sequences) | 344 |
| 345 | (sequences) | 516 |
| 517 | (sequences) | 688 |
| 689 | (sequences) | 860 |
| 861 | (sequences) | 1032 |
| 1033 | (sequences) | 1152 |

**Table 2. Result of BLAST analysis.**

| Sample | Description | Max score | Total score | Query cover | E value | Ident | Accession |
|---|---|---|---|---|---|---|---|
| CnD22166 | *Cervus nippon* isolate NIP1 tRNA-Thr gene, partial sequence; tRNA-Pro gene, D-loop, and tRNA-Phe gene, complete sequence; and 12S ribosomal RNA gene, partial sequence; mitochondrial | 2058 | 2058 | 99% | 0 | 99% | KF141944.1 |
| | *Cervus nippon* isolate NIP2 tRNA-Thr gene, partial sequence; tRNA-Pro gene, D-loop, and tRNA-Phe gene, complete sequence; and 12S ribosomal RNA gene, partial sequence; mitochondrial | 2002 | 2002 | 99% | 0 | 99% | KF141945.1 |
| | *Cervus nippon hortulorum* mitochondrion, complete genome | 1971 | 2057 | 99% | 0 | 99% | HQ191428.1 |
| | *Cervus nippon hortulorum* mitochondrion, complete genome | 1971 | 2057 | 99% | 0 | 99% | GU457433.1 |
| CnD22119 | *Cervus nippon* isolate NIP1 tRNA-Thr gene, partial sequence; tRNA-Pro gene, D-loop, and tRNA-Phe gene, complete sequence; and 12S ribosomal RNA gene, partial sequence; mitochondrial | 1847 | 1847 | 98% | 0 | 96% | KF141944.1 |
| | *Cervus nippon* isolate NIP2 tRNA-Thr gene, partial sequence; tRNA-Pro gene, D-loop, and tRNA-Phe gene, complete sequence; and 12S ribosomal RNA gene, partial sequence; mitochondrial | 1825 | 1825 | 98% | 0 | 96% | KF141945.1 |
| | *Cervus hortulorum* isolate J35D control region, partial sequence; mitochondrial | 1825 | 1825 | 87% | 0 | 99% | JF893529.1 |
| | *Cervus nippon kopschi* mitochondrion, complete genome | 1820 | 1906 | 98% | 0 | 97% | JN389444.1 |
| CnD11102 | *Cervus nippon* isolate NIP1 tRNA-Thr gene, partial sequence; tRNA-Pro gene, D-loop, and tRNA-Phe gene, complete sequence; and 12S ribosomal RNA gene, partial sequence; mitochondrial | 2034 | 2034 | 99% | 0 | 99% | KF141944.1 |
| | *Cervus nippon* isolate NIP2 tRNA-Thr gene, partial sequence; tRNA-Pro gene, D-loop, and tRNA-Phe gene, complete sequence; and 12S ribosomal RNA gene, partial sequence; mitochondrial | 1978 | 1978 | 99% | 0 | 99% | KF141945.1 |
| | *Cervus nippon hortulorum* mitochondrion, complete genome | 1960 | 2033 | 99% | 0 | 99% | HQ191428.1 |
| | *Cervus nippon hortulorum* mitochondrion, complete genome | 1960 | 2033 | 99% | 0 | 99% | GU457433.1 |
| CnD6 | *Cervus nippon* isolate NIP1 tRNA-Thr gene, partial sequence; tRNA-Pro gene, D-loop, and tRNA-Phe gene, complete sequence; and 12S ribosomal RNA gene, partial sequence; mitochondrial | 1847 | 1847 | 99% | 0 | 96% | KF141944.1 |
| | *Cervus hortulorum* isolate J35D control region, partial sequence; mitochondrial | 1831 | 1831 | 88% | 0 | 100% | JF893529.1 |
| | *Cervus nippon* isolate NIP2 tRNA-Thr gene, partial sequence; tRNA-Pro gene, D-loop, and tRNA-Phe gene, complete sequence; and 12S ribosomal RNA gene, partial sequence; mitochondrial | 1825 | 1825 | 99% | 0 | 96% | KF141945.1 |
| | *Cervus hortulorum* isolate J129D control region, partial sequence; mitochondrial | 1825 | 1825 | 88% | 0 | 99% | JF893535.1 |

(*Continued*)

**Table 2.** (Continued)

| Sample | Description | Max score | Total score | Query cover | E value | Ident | Accession |
|---|---|---|---|---|---|---|---|
| CnD8 | *Cervus nippon* isolate NIP1 tRNA-Thr gene, partial sequence; tRNA-Pro gene, D-loop, and tRNA-Phe gene, complete sequence; and 12S ribosomal RNA gene, partial sequence; mitochondrial | 1845 | 1845 | 99% | 0 | 97% | KF141944.1 |
| | *Cervus hortulorum* isolate J35D control region, partial sequence; mitochondrial | 1831 | 1831 | 89% | 0 | 100% | JF893529.1 |
| | *Cervus nippon kopschi* mitochondrion, complete genome | 1825 | 1904 | 99% | 0 | 97% | JN389444.1 |
| | *Cervus nippon sichuanicus* mitochondrion, complete genome | 1825 | 1904 | 99% | 0 | 97% | JN389443.1 |
| WCnD | *Cervus nippon taiouanus* mitochondrion, complete genome | 1954 | 1954 | 97% | 0 | 99% | EF058308.1 |
| | *Cervus nippon taiouanus* mitochondrial DNA, D-loop region and tRNA-Phe, partial sequence, haplotype: 4Twn1 | 1868 | 1868 | 92% | 0 | 99% | AB279722.1 |
| | *Cervus nippon* isolate NIP1 tRNA-Thr gene, partial sequence; tRNA-Pro gene, D-loop, and tRNA-Phe gene, complete sequence; and 12S ribosomal RNA gene, partial sequence; mitochondrial | 1855 | 1855 | 99% | 0 | 97% | KF141944.1 |
| | *Cervus nippon sichuanicus* mitochondrion, complete genome | 1845 | 1845 | 97% | 0 | 98% | JN389443.1 |

showed morphological differentiation stability, and cell differentiation and distribution in the inner cell mass zone were also higher than those of the UEC-non group (Fig 4).

## Discussion

Many deer species are endangered or threatened in their natural habitats [1,31]. The total number of wild deer worldwide is decreasing, and this species is listed as endangered in China. In particular, in South Korea, *Cervus nippon hortulorum* was reported to be extinct, or it is believed that there was a change in the phylogeny due to an exotic species. In particular, studies on the genetic diversity and unique genetic resources of *Cervus nippon hortulorum* species in Korea are limited. According to the results of our study, it is possible that the differences between individuals who inhabited the Korean Peninsula were due to the sika deer living in Russia and adjacent areas, resulting in new subspecies. In 1970, many livestock, including deer, were imported through Korea's livestock improvement project. In addition, in the case of WCnD, which remained in Korea, it is presumed that it was imported from other countries as part of a livestock improvement project. WCnD, a control group of deer thought to have inhabited Korea, was identified as *Cervus nippon taiouanus*, which matched Taiwan's wild species. This indicates that Taiwanese genes were introduced, and CnD22166 and CnD11102 appear similar to the mt-DNA sequence of *Cervus nippon hortulorum*. Among the *Cervus nippon* samples, CnD22166 had 1, 280 bp compared to the 1,000 bp total gene length of WCnD, and had a total haplotype diversity (Hd) of 0.27 and a base diversity (Pi) of 0.015, and had a lower base diversity compared to the sika deer study report (0.021) from Liu et al,. [32]. That is, the sika deer populations in the neighboring areas of the Korean Peninsula showed overall differences in mtDNA structure between the Russian sika deer populations and the Japanese sika deer populations. This study confirmed that the populations in the neighboring areas of the Korean Peninsula could not be clearly distinguished from those in other areas. However, it

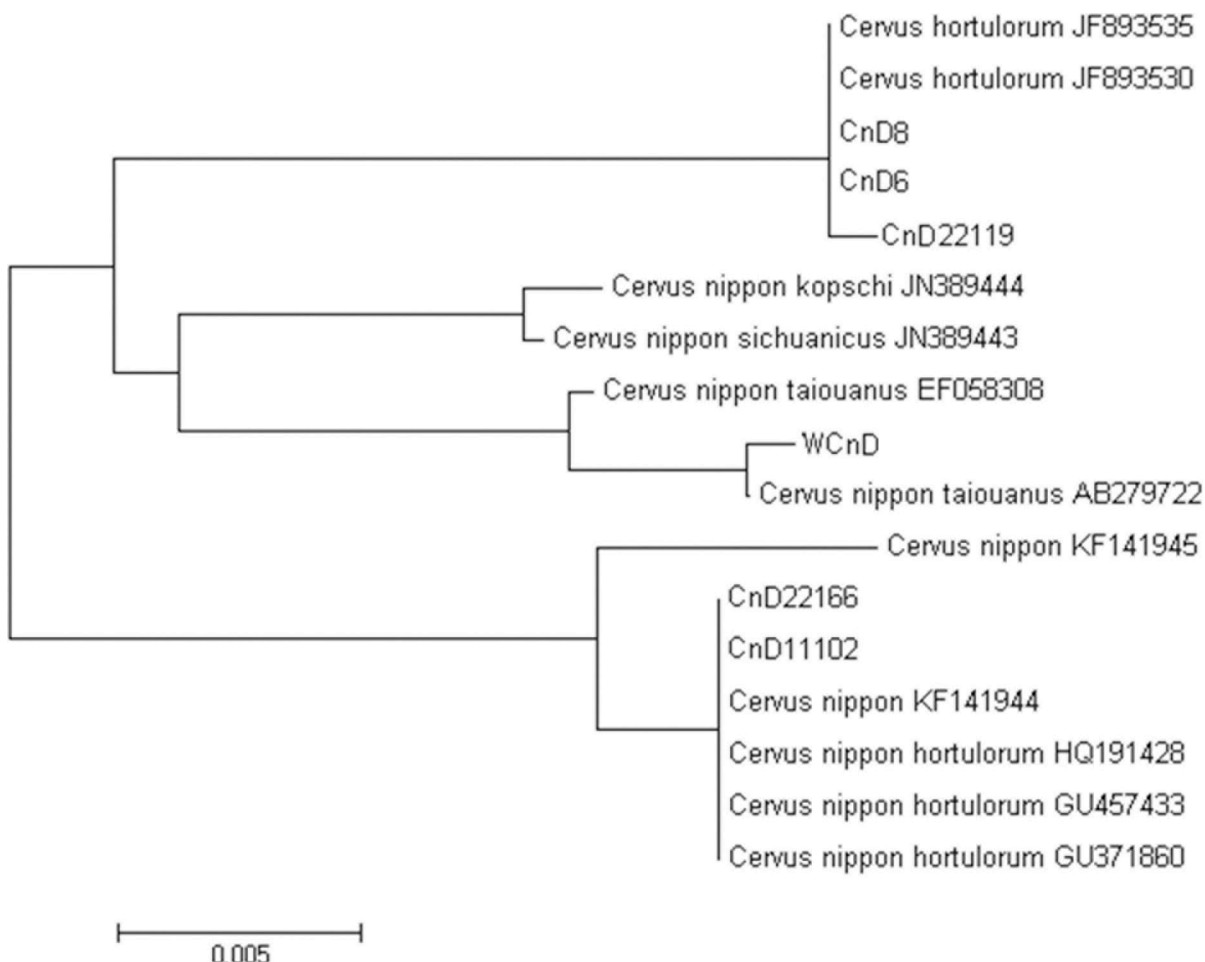

**Fig 2. Phylogenetic analysis of mt-DNA isolated from sika deer based on the neighbor-joining method by MEGA 6.0.**

was confirmed that this species belongs to *Cervus hortulorum* and *Cervus nippon hortulorum*, and is different from *Cervus nippon taiouanus*. In particular, haplotypes of *Cervus nippon hortulorum* and *Cervus nippon taiouanus* were individually grouped into a single branch, which means that classification according to regional differences in the results [32,33] and geographical isolation at a greater distance separates *Cervus nippon taiouanus* from the other three

**Table 3. The embryo development rate of SCNT in porcine oocytes.**

| No. of oocytes used | No. of embryos cleaved | Type of Medium | Percentage (%)[a] of embryos that developed to | | | | |
|---|---|---|---|---|---|---|---|
| | | | 2-cell | 4-cell | 8-cell | Morula | Blastocyst |
| 110 | 30 | NCSU-23 | 25.4 ±0.6 | 13.6 ±0.6* | 12.4±0.4 | 11.8 ±0.3 | 7.9±0.2* |
| 120 | 30 | PZM-3 | 22.4 ±0.5 | 11.6±0.4 | - | - | - |
| 110 | 30 | PZM-5 | 26±0.5* | 13±0.3* | 19.4 ±0.2* | 11.4 ±0.2 | 6.8±0.1 |

[a] Percentage of embryos cultured.

* Different letters within the same column represent significant differences ($p < 0.05$).

A

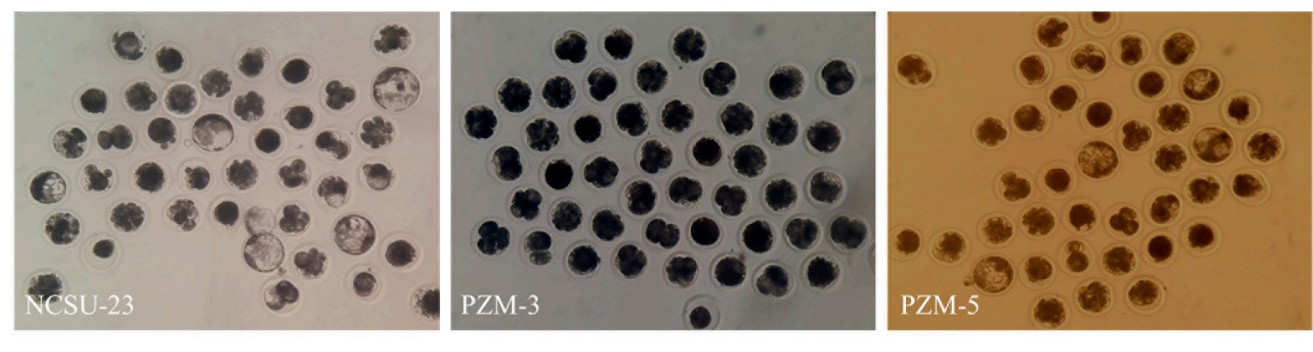

B

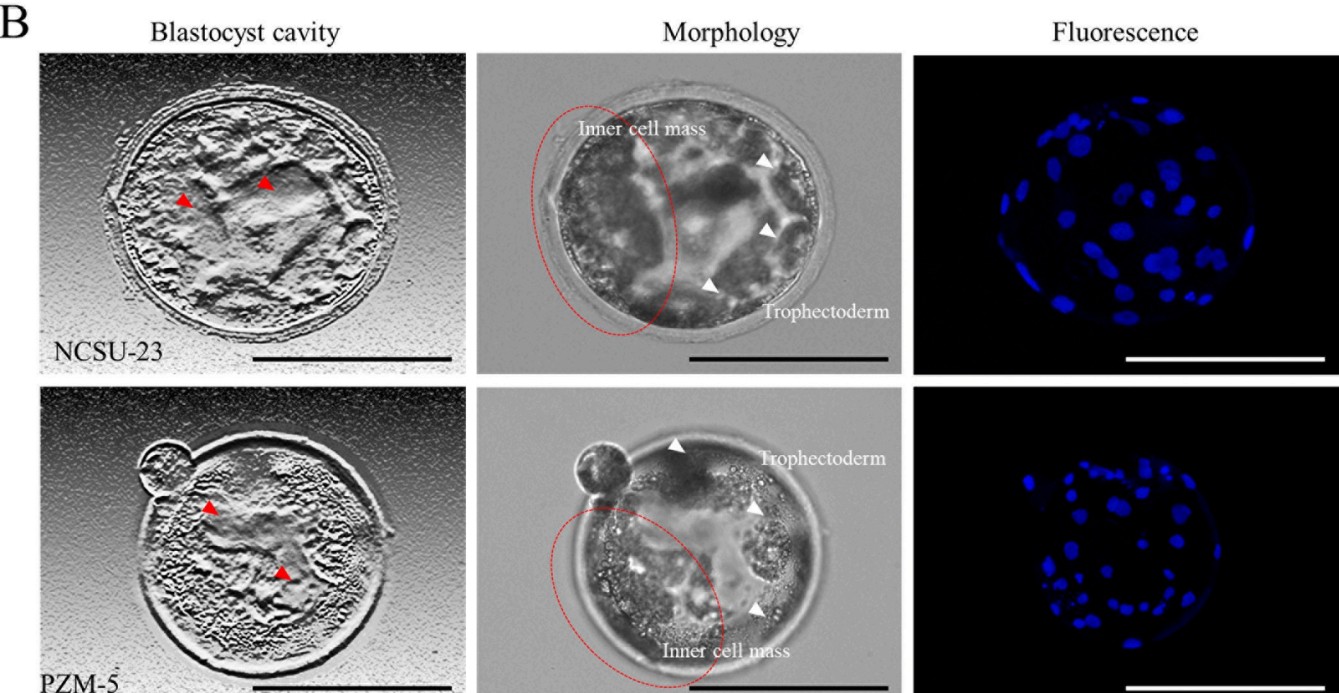

**Fig 3. Development of blastocysts in each culture condition. A** Development of SCNT embryos in each culture medium. **B** The cavity analysis is a photo edited in reverse after phase difference microscopy. The gray shaded photograph was taken by enlarging only one blastocyst. The red arrow points to the blastocyst cavity. The white arrow points to the trophectoderm.

subspecies; thus, there is a low probability of gene exchange in individuals in the neighboring area of the Korean Peninsula due to regional isolation. In many studies, phylogenetic analysis confirmed that there may have been hybridization or secondary contact between the Northeast China and Sichuan populations of *Cervus nippon hortulorum* or incomplete descent classification due to the recent differentiation of the two populations [32,34]. So it is considered that the same affection was also applied to the Korean Peninsula. Therefore, we conducted a study related to the production of nuclear-transfer embryos for species restoration in the future, using cells of sika deer in the neighboring areas of the Korean Peninsula. First, iSCNT was performed on porcine oocytes using frozen CnD22166 and CnD11102 cells, which are very close in genetic distance to *Cervus nippon hortulorum* associated with Taiwan and Russia, which have genetic proximity in mtDNA. According to the study of Ryu et al,. [34], 1.1 kVcm DC pulses were applied for 30 μs to the fusion activation process during the iSCNT process,

**Table 4. The embryo development rate of SCNT in porcine oocytes with co-culture.**

| No. of oocytes used | No. of cleaved embryos | Percentage (%)[a] of embryos that developed to | | | | | | |
|---|---|---|---|---|---|---|---|---|
| | | Co-culture system instead of cell type | | 2-cell | 4-cell | 8-cell | Morula | Blastocyst |
| 110 | 30 | CC | - | 23.9±0.3 | 11.4±0.3 | 9.0±0.4 | 8.4±0.7 | - |
| 110 | 30 | OEC | - | 23.6±0.6 | 13.9±0.6 | 9.6±0.6 | 7.9±0.5 | - |
| 120 | 30 | UEC | - | 22.8±1.8 | 12.4±0.7 | 11.2±0.6 | 9.6±0.4 | 8.2±0.4 |
| 110 | 30 | | LH | 13.9±0.6 | 12.7±0.6 | 8.5±0.2 | 6.9±0.7 | - |
| 120 | 30 | | P4 | 26.0±0.7* | 23.5±0.5* | 13.0±0.7* | 12.4±0.8* | 12.1±0.4* |
| 120 | 30 | | GTH | 23.3±0.7 | 22.7±0.4 | 10.2±0.8 | 7.8±0.5 | - |

[a] Percentage of embryos cultured.

* Different letters within the same column represent significant differences ($p < 0.05$).

CC: Cumulus cell.

OEC: Oviduct epithelial cell.

UEC: Uterine endometrium cell.

confirming that the blastocyst development induction was high. In iSCNT embryo production, the increase of ICM cell homogeneity is very important for improving the quality of blastocysts, and ICM differentiation in interspecific iSCNT is also important for the establishment of embryo stem cells (ESC) [23,35]. In this study, the in vitro culture method for iSCNT embryo production using sika deer cells confirmed that the culture method using NCSU-23 affected the development of blastocysts and the homogeneity of ICM compared to the culture method using PZM [26]. In addition, it was confirmed that the homogeneity of the ICM and blastocyst cavity was sufficient to increase blastocyst metabolism smoothly. As in the study of Lee et al,. [24], owing to the cell death in the section where ICM is formed, it is difficult to secure a sufficient blastocyst cavity, and the homogeneity of ICM can be reduced in the PZM group with low blastocyst cavity. Thus, securing the blastocyst cavity positively affects blastocyst

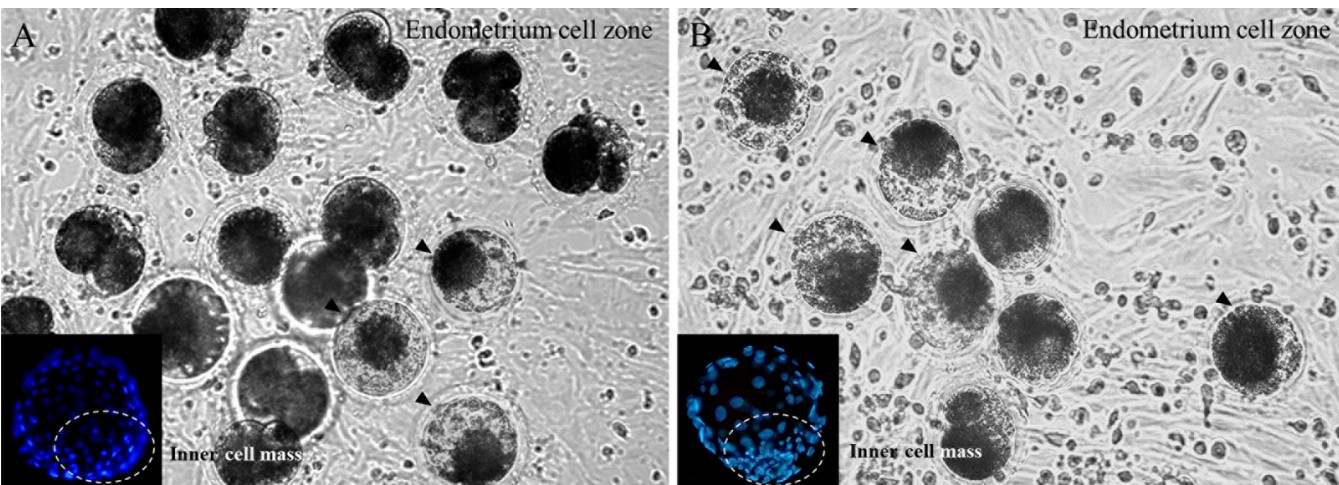

**Fig 4. Development of SCNT-derived embryos in NCSU-23 medium, supplemented A or not supplemented B with hormones (P4 hormones), in co-culture with endometrial cells.** Small picture: Hoechst 33258 staining.

expansion, as shown by Kim et al,. [36–38]. Therefore, in order to secure high-quality blastocysts of iSCNT embryos, the results of the analysis of the occurrence and morphological changes of blastocysts when hormone stimulation was induced during co-culture with uterine cells showed that blastocyst and cell homogeneity increased when inducing progesterone during in vitro culture, and the morphological change of the blastocyst according to the securing of the blastocyst cavity had a positive effect compared to the normal culture. In particular, the characteristics of endometrial cells are thought to induce morphological changes by increasing the metabolism of blastocysts by promoting progesterone-induced development of iSCNT embryos [36–38]. Therefore, studies on the production of iSCNT embryos for the restoration of deer species of the *Cervus nippon hortulorum* lineage on the Korean Peninsula predict that most of the sika deer in the area neighboring the Korean Peninsula belong to *Cervus hortulorum* and *Cervus nippon hortulorum*, but they can be classified as a new subspecies. In addition, it is expected that the production of interspecific nuclear transfer embryos suggests the possibility of species restoration. Many studies are still needed on the production of cloned embryos and animal birth, and ecological characteristics and physiological studies of the restored species remain to be explored; however, there is hope for preservation and restoration of sika deer in Korea, and the use of an in vitro culture method using endometrial cells is thought to have a great influence on the development of stable cloned embryos.

## Conclusion

We analyzed the phylogeny of genes using mt-DNA to determine whether species can be restored by identifying the descent of extinct sika deer in Korea. We confirmed that sika deer in the neighboring areas of the Korean Peninsula could be classified as new subspecies related to the descent of *Cervus nippon hortulorum*. In addition, in producing iSCNT embryos for species restoration, it was possible to suggest a hormone stimulation system for endometrial cells and progesterone to ensure a sufficient blastocyst cavity and positively influence blastocyst formation. Therefore, the results suggest the possibility of restoring extinct sika deer species in Korea. However, studies on the utilization of cloned embryos (iSCNT embryos) and the genetic diversity of Korean sika deer require further analysis. In our study's proposal, management measures for animals born using iSCNT technology still need to be improved. Therefore, creating a representative model of traditional deer is imperative to forming a primary group of deer born with iSCNT technology, observing their long-term adaptability to the ecosystem, and protecting them at the Extinct Animal Restoration Center in Korea.

## Supporting information

**S1 File.**
(DOCX)

## Acknowledgments

We would like to thank Editage (www.editage.co.kr) for English language editing. And, we thank Janggu county (www.jangsu.go.kr) for providing all samples of Sika deer used in this study.

## Author Contributions

**Conceptualization:** Sang-Hwan Kim.

**Data curation:** Yong-Su Park.

**Formal analysis:** Yong-Su Park, Min-Gee Oh, Sang-Hwan Kim.

**Funding acquisition:** Sang-Hwan Kim.

**Investigation:** Min-Gee Oh.

**Methodology:** Yong-Su Park, Min-Gee Oh, Sang-Hwan Kim.

**Project administration:** Sang-Hwan Kim.

**Resources:** Yong-Su Park, Min-Gee Oh.

**Software:** Yong-Su Park, Min-Gee Oh.

**Supervision:** Sang-Hwan Kim.

**Validation:** Min-Gee Oh.

**Visualization:** Yong-Su Park.

**Writing – original draft:** Yong-Su Park, Min-Gee Oh.

**Writing – review & editing:** Yong-Su Park, Sang-Hwan Kim.

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
