## [Decision Letter · Decision Letter 0]

27 Sep 2023

PONE-D-22-34204iSCNT embryo culture system for restoration of Cervus nippon hortulorum, presumed to be sika deer in the Korean PeninsulaPLOS ONE

Dear Dr. Kim,

 Thank you for submitting your manuscript to PLOS ONE. After careful consideration, we feel that it has merit but does not fully meet PLOS ONE’s publication criteria as it currently stands. Therefore, we invite you to submit a revised version of the manuscript that addresses the points raised during the review process. We apologize for the delay in our response. It took us some time to find suitable reviewers. We regret any inconvenience this may have caused.

We look forward to receiving your revised manuscript.

Kind regards,

Birendra Mishra, DVM, PhD

Academic Editor

PLOS ONE

Journal Requirements:

3. Thank you for stating the following in your Competing Interests section: "NO authors have competing interests"

6. Please include your tables as part of your main manuscript and remove the individual files. Please note that supplementary tables (should remain/ be uploaded) as separate ""supporting information"" files. 

**Additional Editor Comments:**

Please upload the histological images with high resolution.

Indicate the specific cells if stained positive.

Authors need to elaborate the findings in a more logical way emphasizing on their won findings along with supportive documents.

Reviewers' comments:

Reviewer's Responses to Questions

**Comments to the Author**

1. Is the manuscript technically sound, and do the data support the conclusions?

Reviewer #1: No

2. Has the statistical analysis been performed appropriately and rigorously? 

Reviewer #1: N/A

3. Have the authors made all data underlying the findings in their manuscript fully available?

Reviewer #1: Yes

4. Is the manuscript presented in an intelligible fashion and written in standard English?

Reviewer #1: No

5. Review Comments to the Author

Reviewer #1: The manuscript titled, “iSCNT embryo culture system for restoration of Cervus nippon hortulorum, presumed to be sika deer in the Korean Peninsula” by Yong-Su Park, Min-Gee Oh and Sang-Hwan Kim described phylogeny of extinct South Korean sika deer and demonstrated the potential of iSCNT for restoration of extinct sika deer. In past century, South Korea has lost several wild species due to habitat loss, war, and uncontrolled hunting. Some species have been restored such as black bear and fox, and still many require efforts to restore. For this, molecular phylogeny is important as it provide information about suitable candidates for reintroduction. As mentioned in the manuscript, the sika still found in Russia Far East and North Korea, thus it can be inferred that sika deer range collapse has happed and it should be reverted by promoting natural or assisted range expansion. iSCNT is excellent for restoring extinct species (the species that are completely wiped out) or critically endangered species where the existing individual number is too low to have impossible (for practical reasons) individual translocations. However, both situations does not fit well with sika deer. There exist wild stable sika deer population in Russia Far East and reintroduced populations in some European countries. Thus efforts should be made through individual capture and relocation via international cooperation. Nevertheless, iSCNT too has future potential and must be studied, discussed and applications to be explored for the wildlife conservation in South Korea.

My comments and queries are as follows:

1. Manuscript require english language editing.

2. Samples – Is WCnD sample collected from South Korea? If yes, please mention sample location and year of collection.

3. A total of six sika deer samples were assed for mitochondrial DNA diversity, of these only one (WCnD) aligned with Cervus nippon taiouanus (Taiwanese sika deer). Rest all showed affinity to Cervus nippon hortulorum. If WCnD is south Korean sika deer, then how researchers concluded Cervus nippon hortulorum as native Korean sika deer. Moreover, the line 203-205 in the result section are unclear and require more explanations as there mentioned about only 3 samples.

4. In the discussion, the author mentioned possible introduction of Taiwanese sika deer in South Korea in the past but the statement lack reference to support. Kindly include.

5. Molecular phylogenetic analysis did not concluded to novel finding. Also, it fail to establish the genetic identity of extinct South Korean sika deer as author discussed the possible source of South Korean wild sika deer been Taiwanese sika deer. Hence, the study need to incorporate more representative samples (if possible) from South Korea.

6. The study lack proper discussion about practical utility and management plan for sika deer restoration in South Korea using iSCNT. The discussion require further modifications and refinement.

6. PLOS authors have the option to publish the peer review history of their article (what does this mean?). If published, this will include your full peer review and any attached files.

Reviewer #1: No

---

## [Author Response · Author response to Decision Letter 0]

10 Feb 2024

Revision Note

▶ All specified items have been modified and marked in red.

Comments from the Editors and Reviewers:

Editor:

Please upload the histological images with high resolution.

Indicate the specific cells if stained positive.

Authors need to elaborate the findings in a more logical way emphasizing on their won findings along with supportive documents.

▶ Thank you very much for the reviewer's comment. We tried to respond diligently based on our opinions.

 

Review Comments to the Author

▶ Thanks for all the comments from the reviewers. We made revisions based on the reviewers' opinions as much as possible.

Reviewer #1 comments

The manuscript titled, “iSCNT embryo culture system for restoration of Cervus nippon hortulorum, presumed to be sika deer in the Korean Peninsula” by Yong-Su Park, Min-Gee Oh and Sang-Hwan Kim described phylogeny of extinct South Korean sika deer and demonstrated the potential of iSCNT for restoration of extinct sika deer. In past century, South Korea has lost several wild species due to habitat loss, war, and uncontrolled hunting. Some species have been restored such as black bear and fox, and still many require efforts to restore. For this, molecular phylogeny is important as it provide information about suitable candidates for reintroduction. As mentioned in the manuscript, the sika still found in Russia Far East and North Korea, thus it can be inferred that sika deer range collapse has happed and it should be reverted by promoting natural or assisted range expansion. iSCNT is excellent for restoring extinct species (the species that are completely wiped out) or critically endangered species where the existing individual number is too low to have impossible (for practical reasons) individual translocations. However, both situations does not fit well with sika deer. There exist wild stable sika deer population in Russia Far East and reintroduced populations in some European countries. Thus efforts should be made through individual capture and relocation via international cooperation. Nevertheless, iSCNT too has future potential and must be studied, discussed and applications to be explored for the wildlife conservation in South Korea.

▶The reviewers' comments and interest are greatly appreciated.

1. Manuscript require english language editing.

▶ English was modified by Editage (www.editage.co.kr).

2. Samples – Is WCnD sample collected from South Korea? If yes, please mention sample location and year of collection.

▶ Samples were collected from the last protected deer in Jangsu County, which died as a roadkill in 2014.

3. A total of six sika deer samples were assed for mitochondrial DNA diversity, of these only one (WCnD) aligned with Cervus nippon taiouanus (Taiwanese sika deer). Rest all showed affinity to Cervus nippon hortulorum. If WCnD is south Korean sika deer, then how researchers concluded Cervus nippon hortulorum as native Korean sika deer. Moreover, the line 203-205 in the result section are unclear and require more explanations as there mentioned about only 3 samples.

▶ At that time, while Korea was importing the deer with which it had diplomatic ties to restore the deer, it was concluded that the WCnD was a native deer because it looked similar to the traditional deer. Of the five samples, only two (CnD11102, CnD22166) were highly probable Korean deer.

4. In the discussion, the author mentioned possible introduction of Taiwanese sika deer in South Korea in the past but the statement lack reference to support. Kindly include.

▶ It is likely that it was introduced to Korea through the livestock improvement project in 1970, but due to the influx of many livestock groups, information on accurate support is insufficient.

The 1970s were a very chaotic time in Korea, both nationally and socially.

5. Molecular phylogenetic analysis did not concluded to novel finding. Also, it fail to establish the genetic identity of extinct South Korean sika deer as author discussed the possible source of South Korean wild sika deer been Taiwanese sika deer. Hence, the study need to incorporate more representative samples (if possible) from South Korea.

▶ It is believed that Korea no longer has native deer. After the death of the only remaining WCnD, representative samples from Korea were no longer available.

6. The study lack proper discussion about practical utility and management plan for sika deer restoration in South Korea using iSCNT. The discussion require further modifications and refinement.

▶ Content was added and some sentences were modified.

* Our research is still negative in Korea. The reason is that because deer are livestock, the importance of the species is low, and animals born with iSCNT technology are not recognized. Although our paper is lacking in many areas, we believe our small results can again spark interest in our country's endemic species.

* Once again, thank you very much for reviewing our research.

---

## [Editor Report · Decision Letter 1]

5 Mar 2024

iSCNT embryo culture system for restoration of Cervus nippon hortulorum, presumed to be sika deer in the Korean Peninsula

PONE-D-22-34204R1

Dear Dr. Park,

We’re pleased to inform you that your manuscript has been judged scientifically suitable for publication and will be formally accepted for publication once it meets all outstanding technical requirements.

Kind regards,

Birendra Mishra, DVM, PhD

Academic Editor

PLOS ONE

Additional Editor Comments (optional):

Thanks for responding the reviewers comments.
---

## [Editor Report · Acceptance letter]

4 Apr 2024

PONE-D-22-34204R1 

PLOS ONE

Dear Dr. Kim, 

I'm pleased to inform you that your manuscript has been deemed suitable for publication in PLOS ONE. Congratulations! Your manuscript is now being handed over to our production team.

Kind regards, 

on behalf of

Dr. Birendra Mishra 

Academic Editor

PLOS ONE